# Leaf wax *n*-alkanes in modern plants and topsoils from eastern Georgia (Caucasus) – implications for reconstructing regional paleovegetation

Marcel Bliedtner[1,2], Imke K. Schäfer[2], Roland Zech[1,2], Hans von Suchodoletz[3]

[1]Institute of Geography, University of Jena, Löbdergraben 32, D-07743 Jena, Germany
[2]Institute of Geography and Oeschger Centre for Climate Change Research, University of Bern, Hallerstrasse 12, CH-3012 Bern, Switzerland
[3]Institute of Geography, University of Leipzig, Johannisallee 19a, D-04103 Leipzig, Germany

*Correspondence to*: Marcel Bliedtner (marcel.bliedtner@uni-jena.de)

**Abstract.**

Long-chain *n*-alkanes originate from leaf waxes of higher terrestrial plants, are relatively resistant against physical and chemical degradation and are preserved in sediment archives for at least millennial timescales. Since their homologue patterns discriminate between vegetation forms, they were increasingly used for paleovegetation reconstructions during the last years. However, before any robust interpretation of the long-chain *n*-alkane patterns in sediment archives, reference samples from modern vegetation and topsoil material should be investigated at a regional scale. Up to now, such systematic regional studies on modern plant and topsoil material exist for temperate and tropical regions but are still largely lacking for semi-humid to semi-arid regions.

To test the potential of leaf wax-derived *n*-alkane patterns for paleoenvironmental studies in the semi-humid to semi-arid central southern Caucasus region, we investigated the influence of different vegetation forms on the leaf wax *n*-alkane signal in modern plants and topsoil material (0-5 cm) from eastern Georgia. We sampled (i) sites with grassland/herbs that included steppe, cultivated grassland and meadows, and (ii) sites that are dominated by deciduous hornbeam forests.

The results show that long-chain *n*-alkanes originate from leaf waxes of higher terrestrial plants and that their homologue pattern allow to discriminate between different vegetation forms: *n*-Alkanes derived from sites with grassland/herbs are mainly dominated by $C_{31}$, while *n*-alkanes derived from sites with deciduous trees/shrubs show high abundances of $C_{29}$. Thus, long-chain *n*-alkanes have a great potential when used for regional paleovegetation reconstructions. Moreover, the *n*-alkane distributions of the topsoils do not show correlations with mean annual temperatures and precipitation along the investigated transect. As degradation of organic matter can affect the leaf wax *n*-alkane distribution, we further present an updated end-member model that includes our results, accounts for degradation effects and enables semi-quantitative reconstructions of past vegetation changes in the central southern Caucasus region.

# 1 Introduction

Long-chain $n$-alkanes ($C_{25}$-$C_{35}$) are produced as part of the epicuticular leaf waxes by terrestrial plants and can thus serve as valuable biomarkers (Eglinton et al., 1962; Eglinton and Hamilton, 1967; Kolattukudy and Walton, 1973). Typically, leaf wax $n$-alkanes show a distinct odd-over-even predominance (OEP) (Eglinton and Hamilton, 1967), and their relative odd homologue distribution might be used to differentiate between different vegetation forms: The chain-lengths $C_{27}$ and $C_{29}$ are mainly produced by deciduous trees and shrubs, while $C_{31}$ and $C_{33}$ mainly derive from grasses and herbs (Marseille et al., 1999; Zech et al., 2009; Schwark et al., 2002). Because of their low water-solubility and relative persistence against physical and chemical degradation, they stay well preserved in soils and sedimentary archives at least over millennial timescales (Eglinton and Eglinton, 2008). Up to now, long-chain $n$-alkanes have increasingly been applied to various paleoenvironmental archives such as lacustrine and marine sediments and loess-paleosol sequences to reconstruct paleovegetation (Schwark et al., 2002; Schefuß et al., 2003; Zhang et al., 2006; Liu and Huang, 2005; Schatz et al., 2011; Zech et al., 2009; Schäfer et al., 2016a).

However, robust interpretations of long-chain $n$-alkanes in sedimentary archives can only be obtained by regional calibration studies on modern plant and topsoil material. The need for regional calibration studies is underlined by the study of Bush and McInerney (2013) that report no discrimination of modern vegetation forms by $n$-alkanes from different sites around the world, and therefore they questioned whether $n$-alkane patterns are suitable to distinguish between different vegetation forms on a global scale. In contrast, although still lacking for many regions, some regional studies from Central and South-Eastern Europe and tropical South America and Africa demonstrate the discrimination power of $n$-alkanes for vegetation reconstructions, although the most abundant $n$-alkane homologues show regional differences (Zech et al., 2009; Zech et al., 2013; Schäfer et al., 2016b; Feakins et al., 2016; Vogts et al., 2009): While Schäfer et al. (2016b) report $C_{27}$ as the most abundant homologue produced by deciduous trees/shrub vegetation in humid Central Europe, Zech et al. (2013) show that $C_{29}$ is the dominant homologue produced by this vegetation form in more arid South-Eastern Europe. In tropical ecosystems, $C_{29}$ is the most abundant homologue in C3 rainforest trees, whereas $C_{33}$ is mainly produced by C4 grasses and herbs (Vogts et al., 2009; Feakins et al., 2016). Moreover, several potential pitfalls can complicate the interpretation of long-chain $n$-alkanes, as there are relatively wide species-specific variations of the produced $n$-alkanes (Diefendorf et al., 2011; Bush and McInerney, 2013), environmental factors (Hoffmann et al., 2013; Tipple and Pagani, 2013) and degradation effects (Buggle et al., 2010) that can potentially influence the $n$-alkane pattern. Therefore, prior to the application of long-chain $n$-alkanes for paleovegetation reconstructions in sedimentary archives, regional calibration sets from modern long-chain $n$-alkane patterns are necessary to assess their discrimination power between grasses/herbs and deciduous trees/shrubs.

This study evaluates long-chain $n$-alkane patterns of modern plant and topsoil material from eastern Georgia in the central southern Caucasus region to test their chemotaxonomic potential, i.e. whether $n$-alkane abundances and chain-length distributions can be used to discriminate between deciduous trees/shrubs and grasses/herbs on a regional level. Since climatic factors can potentially influence the $n$-alkane pattern, we checked such an influence by correlating mean annual temperature and precipitation with the $n$-alkane distribution. Furthermore, the leaf wax $n$-alkane pattern can be affected by degradation effects,

but these can be checked and corrected for with regionally derived end-members (Zech et al., 2013; Schäfer et al., 2016b). Thus, we further aim to establish a regional end-member model of *n*-alkanes for the central southern Caucasus region as a base for future robust interpretations of leaf wax *n*-alkane patterns derived from regional sediment archives. Such lacustrine, palustrine, fluvial, loess-paleosol or open-air archaeological sediment archives were increasingly investigated during the last years and could potentially be used for vegetation reconstructions based on *n*-alkanes (Messager et al., 2013; Joannin et al., 2014; Suchodoletz et al., 2015; Wolf et al., 2016; Egeland et al., 2016).

## 2 Material and Methods

### 2.1 Geographical setting and field sampling

The semi-humid to semi-arid central southern Caucasus is characterized by a small-scale pattern of different ecologic regions. Due to a rain-shadow that is caused by the high elevation of the Western and Central ranges of the Greater Caucasus and by the NW-SE-directed Likhi Range linking the Greater and Lesser Caucasus (Fig. 1), there exists a significant climatic gradient with decreasing precipitation and more continental conditions going eastwards (Connor and Kvavadze, 2008). Our study of modern plant and topsoil material was carried out in eastern Georgia (see Fig. 1) at the end of the vegetation period in September 2016. We sampled 22 sites that are located along a N-S-directed transect with a length of ca. 65 km. The transect extends from the surroundings of Tbilisi in the south over the mid-mountain Kura-fold-and-thrust-belt (Forte et al., 2010) into the upper part of the Alazani valley in the north. The sampled sites with grassland/herbs included steppe, cultivated grassland and meadows, and the sites with deciduous trees/shrubs were mainly dominated by hornbeam forests (see Fig. 2A for sample locations and supporting online material SOM-1 and 2 for site descriptions). At each site, we sampled plant material and the upper 0-5 cm of the topsoil.

The altitudes along the transect range between 445 and 1659 m a.s.l., mean annual temperatures between ca. 5.4 and 13.3°C (www.yr.no/place/Georgia/Tbilisi; see chapter 2.3) and mean annual precipitation between ca. 600 and 2000 mm (unpublished precipitation map of W. Bagrationi Geographical Institute Tbilisi; Fig. 2C). Precipitation mainly falls in spring and early summer during convective events (Lydolph, 1977). The recent vegetation of eastern Georgia is part of the Irano–Turanian Group (Connor et al., 2004; Sagheb-Talebi et al., 2014). In the surroundings of Tbilisi, the natural vegetation is characterized by xerophytes and semidesert vegetation, while the vegetation on the southern slopes of the mid-mountain Kura-fold-and-thrust-belt changes from oak-hornbeam forests in the lower parts to mixed beech forests in the upper parts. In the semi-humid lowlands of the upper Alazani valley, the natural vegetation can be classified as elm-oak-vine forests. However, agricultural fields cover most of the valley floor today. The mid-mountain belt of the upper Alazani valley is characterized by mixed beech and hornbeam forests and in small parts also by fir-spruce forests (Connor and Kvavadze, 2008, Fig. 2B).

### 2.2 Leaf wax analyses

**Analytical procedure**

Leaf waxes were analysed at the Institute of Geography of the University of Bern/Switzerland. Free lipids from modern plant and topsoil samples were extracted using an ultrasonic treatment. Plant material (~1g) and soil material (~10g) were extracted three times with 20 ml dichloromethane (DCM): methanol (MeOH) (9:1, *v/v*) in an ultrasonic bath for 15 min.

The total lipid extract was separated over aminopropyl pipette columns into: i) the apolar fraction including the *n*-alkanes, ii) the polar fraction, and iii) the acid fraction. The *n*-alkanes were eluted with ~4 ml hexane and further purified over coupled silver-nitrate (AgNO$_3$-) - zeolite pipette columns. The *n*-alkanes trapped in the zeolite were subsequently dissolved in hydrofluoric acid and recovered by liquid-liquid extraction using *n*-hexane.

Afterwards, the *n*-alkanes were identified and quantified using a gas chromatograph with a flame ionisation detector (GC-FID). GC-FID measurements were performed on an Agilent 7890 gas-chromatograph equipped with an Agilent HP5MS column (30 m * 320 μm * 0.25 μm film thickness). For quantification and identification, external *n*-alkane standards ($C_{21} - C_{40}$) were run with each sequence.

**Data analyses**

*n*-Alkane concentrations were calculated as the sum of $C_{25}$ and $C_{35}$, and given in μg g$^{-1}$ dry weight.

Odd-over-even predominance (OEP) values (Eq. 1) were determined following Hoefs et al. (2002). Low values (<5) indicate an enhanced state of degradation (Buggle et al., 2010; Zech et al., 2009).

$$OEP = \frac{C27+C29+C31+C33}{C26+C28+C30+C32} \tag{1}$$

The average chain length (ACL) of *n*-alkanes (Eq. 2) was calculated after Poynter et al. (1989), and was used to distinguish between leaf waxes that predominantly originate from deciduous trees and shrubs ($C_{27}$ and $C_{29}$; i.e. lower values) and those mainly originating from grass/herb vegetation ($C_{31}$ and $C_{33}$; i.e. higher values).

$$ACL = \frac{27*C27+29*C29+31*C31+33*C33}{C27+C29+C31+C33} \tag{2}$$

The *n*-alkane ratio that is used for the end-member model to correct for degradation effects (Zech et al., 2013) is a normalized ratio to differentiate between vegetation forms, with higher values for grasses and herbs and lower values for deciduous trees and shrubs (Eq. 3).

$$n - alkane\ ratio = \frac{C31+C33}{C27+C29+C31+C33} \tag{3}$$

**2.3 Climatic parameters**

Climatic parameters, namely mean annual temperatures and precipitation, were correlated with the *n*-alkane ratio to test potential climatic influences on the *n*-alkane distribution. Generally, climate data for eastern Georgia are scarce because only few climate stations are located within the study area, and almost all stations show large data gaps after 1990 due to the political problems following the decay of the Soviet Union. Thus, reliable climate data cannot be derived from climate stations along

the investigated transect or even for each sampled site. To overcome these limitations, we used mean annual temperature data of the climate station Tbilisi ([www.yr.no/place/Georgia/Tbilisi](www.yr.no/place/Georgia/Tbilisi)) that is located at 490 m a.s.l. and reports temperatures between 1962 and 1990. Based on this reference station, mean annual temperatures for our investigated sites were calculated under the assumption that temperature declines linearly with 0.65 k by 100 m increase in elevation (barometric formula of Schönwiese, 2008). The approximate altitude of each site was taken by hand-held GPS. For precipitation, we used an unpublished precipitation map of Georgia (V. Bagrationi Geographical Institute Tbilisi) that reports extrapolated mean annual precipitation between 1960 and 1990 (Fig. 2C). Since the available climate data average over ca. 30 years, only the $n$-alkane distributions from the topsoils representing averaged decadal signals (Wiesenberg et al., 2004) can robustly be correlated with the climate data. In contrast, the $n$-alkane distributions from the corresponding modern plants represent annual signals of the sampling year 2016, and can thus not robustly be correlated with the available climate data but will be shown for completeness.

## 3 Results

$n$-Alkanes are present in all analysed modern plant and topsoil samples from eastern Georgia, and the $n$-alkane and climate data are provided in the *supporting online material SOM-3*.

All samples show a distinct dominance of odd-numbered long-chained $n$-alkanes ($>C_{25}$). The total $n$-alkane concentrations ($C_{25}$-$C_{35}$) of the samples range from 1.1 to 299.1 µg g$^{-1}$ dry material, and modern plant samples have systematically higher $n$-alkane concentrations than topsoil samples (Fig. 3a). All samples show distinct odd-over-even predominance (OEP) values between 3.1 and 18.7, and especially for samples from sites with grassland/herbs, modern plant material shows systematically higher OEP values than the corresponding topsoil material (Fig. 3b). Average-chain-length (ACL) values range from 28.6 to 31.2, with samples from sites with grassland/herbs showing systematically higher values than those from sites with deciduous trees/shrubs (Fig. 3c). Systematic differences in chain-length patterns are further reported by the $n$-alkane ratio that ranges from 0.16 to 0.77 (Fig. 3d): Lower $n$-alkane ratios ($<0.5$) are characteristic for the samples from sites with deciduous trees/shrubs, whereas higher ratios ($>0.5$) are typical for samples from sites with grassland/herbs. Likewise, systematic differences in the chain-length patterns between samples from sites with deciduous trees/shrubs and sites with grassland/herbs are illustrated in Fig. 4: Samples from sites with deciduous trees/shrubs show a clear dominance of $C_{29}$, whereas samples from sites with grassland/herbs are dominated by $C_{31}$. This pattern is characteristic for both, analysed modern plant and topsoil material, although the standard deviation of $C_{29}$ and $C_{31}$ of modern plant samples is generally wider, what holds especially true for sites with grassland/herbs (Fig. 4).

$n$-Alkanes from modern plants and topsoils show no correlation with mean annual temperatures for both, $n$-alkanes from sites with grassland/herbs ($R^2 = 0.0873$ for modern plants; $R^2 = 0.0947$ for topsoils) and from sites with deciduous trees/shrubs ($R^2 = 0.0254$ for modern plants; $R^2 = 0.1855$ for topsoils) (Fig. 5a). With respect to mean annual precipitation, no correlation exists for all $n$-alkanes from sites with grassland/herbs ($R^2 = 0.0001$ for modern plants; $R^2 = 0.0621$ for topsoils), and $n$-alkanes from topsoils from sites with deciduous trees/shrubs ($R^2 = 0.0085$). Only the $n$-alkanes from modern plants from sites with deciduous trees/shrubs show a weak correlation with mean annual precipitation ($R^2 = 0.3362$) (Fig. 5b).

# 4 Discussion

## 4.1 *n*-Alkane patterns in modern plants and topsoils

All modern plant and topsoil samples from eastern Georgia show a distinct OEP. This proves their leaf wax origin (Eglinton and Hamilton, 1967). Most fresh plant material shows higher *n*-alkane concentrations and OEPs compared with topsoils (Fig. 3a, b), what indicates ongoing degradation of organic matter (OM) in the topsoils (Buggle et al., 2010; Schäfer et al., 2016b). However, also in the topsoils the OEP values are mostly >5. This indicates still good preservation of the *n*-alkanes, allowing the reconstruction of former vegetation patterns. Accordingly, both the ACL and the *n*-alkane ratio report distinct differences in the chain-length patterns for modern plant and topsoil samples from sites with deciduous trees/shrubs and sites with grassland/herbs (Figs. 3, 4): Lower ACL values and a dominance of $C_{29}$ are characteristic for samples from sites with deciduous trees/shrubs, and higher ACL values and a $C_{31}$ dominance for *n*-alkanes from sites with grassland/herbs (Fig. 3c, d). However, modern plant samples 9p, 25p and 34p originating from sites with grassland/herbs show a chain-length dominance of $C_{29}$, whereas modern plant samples 3p, 5p and 29p and topsoil sample 23s originating from sites with deciduous trees are dominated by $C_{31}$ homologues. Thus, these samples do not support the proposed chain-length patterns for the respective vegetation forms. At the sampling site of topsoil sample 23s, hornbeam trees were only growing in patches and the site is and probably was intensively used for grazing activities. Likewise, Holtvoeth et al. (2016) report similar *n*-alkane distributions from the Ohrid Basin, i.e. a dominance of grass-derived ($C_{31}$) *n*-alkanes in the topsoil of a beech forest. Besides vegetation changes, they suggest that the *n*-alkanes from the grassy undergrowth overproportionally enter the soil, whereas the leaf litter is more mobile and might thus get relocated. Therefore, the dominance of $C_{31}$ in the topsoil of site 23 might be an inherited signal from former land-use/vegetation change, and/or because *n*-alkanes from the grassy undergrowth were overproportionally incorporated into the topsoil. For the observed outliers of the modern plant samples, it is possible that these reflect: (i) a relatively large species-specific variability of the produced *n*-alkane pattern that was repeatedly reported in the literature (Bush and McInerney, 2013; Diefendorf et al., 2011) and/or (ii) an influence of annual temperature (Tipple and Pagani, 2013; Bush and McInerney, 2015; Sachse et al., 2006) and moisture variations (Hoffmann et al., 2013) on the homologue patterns. Modern grassland/herb sample 9 with $C_{29}$ dominance was taken from a site in higher altitudes of the Kura-fold-and-thrust-belt (1659 m a.s.l.) that could potentially be affected by extreme temperatures. Therefore, this factor might potentially have affected the *n*-alkane synthesis at this site during the growing season of 2016. In contrast, modern grassland/herb samples 25 and 34 with $C_{29}$ dominance were taken from sites located at an inactive former floodplain of the Alazani River in the upper Alazani Basin at altitudes of 570 and 445 m a.s.l., respectively (Suchodoletz et al., in press). Thus, they are not expected to be affected by temperature or moisture extremes. Similarly, modern deciduous tree/shrub samples 3 (924 m a.s.l.) and 5 (830 m a.s.l.) with a strong contribution of $C_{31}$, and modern deciduous tree/shrub sample 29 (774 m a.s.l.) with a dominance of $C_{31}$ were not taken from sites where an influence of temperature or moisture extremes should be expected. Therefore, given that the respective topsoil samples from these sites that average the *n*-alkane patterns over several years/decades (Wiesenberg et al., 2004) show the expected $C_{31}$ dominance for grassland sites 9, 25 and 34, and the expected $C_{29}$ dominance for deciduous tree/shrub sites 3,

5 and 29, an influence of interannual temperature or moisture variations on the observed outliers of the $n$-alkane patterns in 2016 can only be assumed so far. A generally higher sensitivity of our modern plant samples towards interannual variations of the $n$-alkane synthesis is also supported by generally larger standard deviations of the $n$-alkane parameters from modern plant material (Figs. 3 and 4).

To get more insight into possible climatic influences on the $n$-alkane distributions in eastern Georgia, we correlated our $n$-alkane ratios with mean annual temperatures and precipitation (Fig. 5). The only available climate information for our investigated transect are averaged temperature and precipitation data for the timespan between 1960 and 1990. These can only be correlated with the $n$-alkane signals from topsoils, since these are averaged over a similar decadal timescale (Wiesenberg et al., 2004). As shown in Fig. 5, the $n$-alkanes from topsoils do neither correlate with mean annual temperatures nor with pre-

cipitation. The weak correlation between the $n$-alkane pattern of modern plants from sites with deciduous trees/shrubs for the sampling year 2016 and mean annual precipitation between 1960 and 1990 (Fig. 5b) should not be over-interpreted due to the different periods that are covered by both datasets. Summing up, a significant influence of climatic factors on the $n$-alkane patterns in eastern Georgia could not be observed. These findings seem to contradict former studies that repeatedly found correlations between the $n$-alkane distribution and temperature (Tipple et al., 2013; Bush and McInerney, 2015) or precipitation

(Hoffmann et al., 2013). However, those studies investigated climatic transects with wide temperature gradients of >20 °C and precipitation gradients spanning ~2000 mm of annual rainfall. In contrast, our investigated transect shows much smaller gradients in temperature and precipitation of 7.9 °C and ca. 1400 mm/a, respectively. Therefore, climate has obviously no significant influence on the $n$-alkane distribution in topsoils of eastern Georgia. Since climate seems a negligible factor for the $n$-alkane distribution in topsoils, this further justifies our approach to reconstruct regional paleovegetation based on $n$-alkane

patterns at least in eastern Georgia.

    Taken together, although it has recently been questioned whether $n$-alkane patterns originate from leaf waxes and are suitable to distinguish between different vegetation forms at all (Bush and McInerney, 2013), our results from eastern Georgia confirm the leaf wax origin and chemotaxonomic discrimination power of the $n$-alkane pattern from modern plant and topsoil samples for this region. This agrees with previous studies from South-Eastern Europe where $C_{29}$ was reported as the most abundant

homologue derived from deciduous trees and shrubs, and $C_{31}$ as mainly derived from grasses and herbs (Zech et al., 2009; Holtvoeth et al., 2016). A similar good chemotaxonomic discrimination power of the $n$-alkane pattern has also been reported from other regions, although slight differences in the patterns were observed between the individual regions: Schäfer et al. (2016b) found that high amounts of $C_{31}$ and $C_{33}$ were produced at grassland/herb sites in temperate Central Europe, and that sites with deciduous forests/shrubs were mostly dominated by $C_{27}$. In tropical ecosystems, several studies observed $C_{29}$ as the

most dominant $n$-alkane homologue in C3 rainforest trees, whereas C4 grasses and herbs mainly produced $C_{33}$ as the most abundant homologue (Feakins et al., 2016; Vogts et al., 2009; Garcin et al., 2014). For more arid South-Africa, a dominance of $C_{31}$ and $C_{33}$ $n$-alkanes is reported for succulent plants and regional soils (Carr et al., 2014). Altogether, most regional studies show a good chemotaxonomic discrimination power of $n$-alkane patterns, demonstrating their great potential to reconstruct paleovegetation when used in sedimentary archives at least in these regions.

## 4.2 Implications for reconstructing paleovegetation composition

When using leaf wax *n*-alkanes in sedimentary archives, degradation of OM needs to be taken into account since this process leads to preferential losses of the most abundant homologues. Changes of the homologue distribution and potentially confounding degradation effects can be visualized in an end-member model (Zech et al., 2009; Zech et al., 2013) that is based on

modern plant/litter and topsoil samples from sites with deciduous trees/shrubs and sites with grassland/herbs in Central and South-Eastern Europe. This model uses the normalized *n*-alkane ratio $(C_{31}+C_{33})/(C_{27}+C_{29}+C_{31}+C_{33})$ on the y-axis and the OEP on the x-axis: Whereas the *n*-alkane ratio discriminates between deciduous trees/shrubs and deciduous forest soils that show lower values on the one hand, and grasses/herbs and soils from grassland/herb sites that show higher values on the other hand, the OEP is used as a proxy for degradation. The respective trend lines, referred to as "degradation lines" for grass/herb and

deciduous tree/shrub vegetation, illustrate how the *n*-alkane ratio changes with degradation (Fig. 6): With increasing degradation, the degradation lines of grasses and deciduous trees converge, and therefore the discrimination power of the *n*-alkane ratio systematically reduces with increasing degradation. Given that most modern plant and topsoil *n*-alkanes from eastern Georgia discriminate well between sites with deciduous trees/shrubs and sites with grassland/herbs when using the *n*-alkane ratio of Zech et al. (2013; Eq. 3), we integrated our *n*-alkane samples from eastern Georgia into that end member model to

enhance the data base and thus to better back the model for eastern Georgia (Fig 6): Whereas most samples from eastern Georgia plot well along the respective degradation lines of grasses/herbs and deciduous trees/shrubs, this is not the case for modern plant samples 3p, 5p, 9p, 25p, 29p and 34p and topsoil sample 23s. For the latter sample this was possibly caused by former land use/vegetation changes and/or overproportional incorporation of grass-derived *n*-alkanes into the topsoil, and for plant sample 9p possibly by strong interannual temperature variations, whereas for the remaining samples the causes still

remain unclear (Fig. 6, see chapter 4.1). All these samples were regarded as outliers and thus not considered when recalculating the degradation lines after integration of the samples from eastern Georgia. Given that all but one topsoil sample that average the *n*-alkane signal over several decades show a very good discriminative power between grass/herb and deciduous tree/shrub vegetation, this demonstrates the robustness of the *n*-alkane signal as a proxy for paleovegetation reconstructions at least in eastern Georgia (Fig. 6). The recalculated degradation lines for the relative contributions of grasses/herbs and deciduous

trees/shrubs can now serve as a base to calculate former vegetation distributions from paleosamples that were taken from sedimentary archives in eastern Georgia and neighbouring regions of the central southern Caucasus: Depending on the vegetation form that contributed most to the OM of a sample, the *n*-alkanes will plot in the two-component mixing end-member model on or near to one of the degradation lines of grasses or deciduous trees, respectively. The OEP lowers with increasing degradation. Therefore, increasing degradation reduces the accuracy of the correction, since the degradation lines converge

with lowered OEPs. The percentage of grasses/herbs can be calculated using the following formula:

$$\% \text{ grass} = \frac{n-\text{alkane ratio (sample)} - \text{equation (degradation line trees)}}{\text{equation (degradation lines grass)} - \text{equation (degradation lines trees)}} * 100 \qquad (4)$$

Although the OEP should generally decrease when plants are degraded and incorporated into soil organic matter, samples 16p/16s, 20p/20s, 23p/23s and 35p/35s originating from sites with deciduous trees/shrubs do not show the expected pattern (Fig. 6). However, the differences in the OEP values from those sites are only minor (i.e. <1) and thus negligible, but the degradation of leaf wax *n*-alkanes in soils and sediments is not fully understood so far.

5 The observed scattering of the individual samples around the degradation lines, potentially leading to negative or >100% values when calculating grass/herb percentages, could be caused by species-specific and environmental variability: Even within one species, environmental stress like different temperatures, moisture conditions, radiation levels, shading or soil nutrient availability can cause some variation of the *n*-alkane synthesis (Shepherd and Wynne Griffiths, 2006). Given that our *n*-alkane topsoil samples do not show correlations with temperature and annual precipitation (see chapter 4.1; Fig. 5), these 10 factors can possibly be ruled out as causes for the observed scatter of topsoil *n*-alkanes in eastern Georgia.

## 5 Conclusions

This study systematically investigates long-chain *n*-alkanes in modern plant and topsoil material from eastern Georgia in the central southern Caucasus to test their potential for paleovegetational studies. Our results illustrate that:

    i.)    Long-chain *n*-alkanes ($C_{25}$-$C_{35}$) show a clear OEP and thus originate from the leaf waxes of higher terrestrial plants.
15     The regional leaf wax *n*-alkane patterns show distinct and consistent differences between samples from sites with grassland/herbs and sites with deciduous trees/shrubs: Samples from grassland/herb sites are characterized by a dominance of the homologue $C_{31}$, whereas samples from sites with deciduous trees/shrubs show higher abundances of $C_{29}$. Thus, leaf wax *n*-alkanes that discriminate between these two vegetation forms have a great potential for paleovegetation reconstructions in the semi-humid to semi-arid central southern Caucasus region. Besides vegetation 20     changes, also other factors such as interannual temperature and moisture variability obviously influenced the chain-length abundances of some modern plant samples, and factors such as land use/vegetation change or varying incorporation of OM from different plant types the chain-length abundances of one topsoil sample. These were only second-order effects at our investigated sites and accordingly, the *n*-alkane distributions of the topsoils did not show correlations with the climatic parameters annual temperature and precipitation along the investigated transect.

25     ii.)    Given that increasing degradation reduces the vegetation-specific differences in the *n*-alkane homologue patterns, the degradation state of the *n*-alkanes has to be taken into account. Therefore, to allow for a better correction of regional paleovegetation reconstructions from paleosamples we suggest to integrate our plant and topsoil samples from eastern Georgia into the end-member model of Zech et al. (2013) that corrects for these effects. Generally, in case that the *n*-alkane patterns are very similar to those that were used for the original endmember model of Zech et al. (2013) as is 30     the case for eastern Georgia, the accuracy of the correction can be increased by integrating more regional reference samples into the model. However, in case of *n*-alkane patterns that significantly deviate from those that were used for the endmember model of Zech et al (2013), the construction of an independent regional endmember model would be necessary.

Overall, our findings are in good agreement with other regional studies from Central and South-Eastern Europe and demonstrate the high potential of leaf wax *n*-alkanes for paleovegetation reconstruction. However, regional calibration studies with modern reference samples are a necessary base for the robust interpretation of leaf wax-derived long-chain *n*-alkanes in terms of former vegetation changes.

## 6 Data availability

The dataset that is used in this study is available via the supporting online material.

**Acknowledgements**

We thank Ulrich Göres (Dresden) and Giorgi Merebashvili (Tblisi) for their help during fieldwork. We thank Tobias Stalder (Bern) for laboratory work. This project was financially supported by the Swiss National Science Foundation (PP00P2-150590).

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

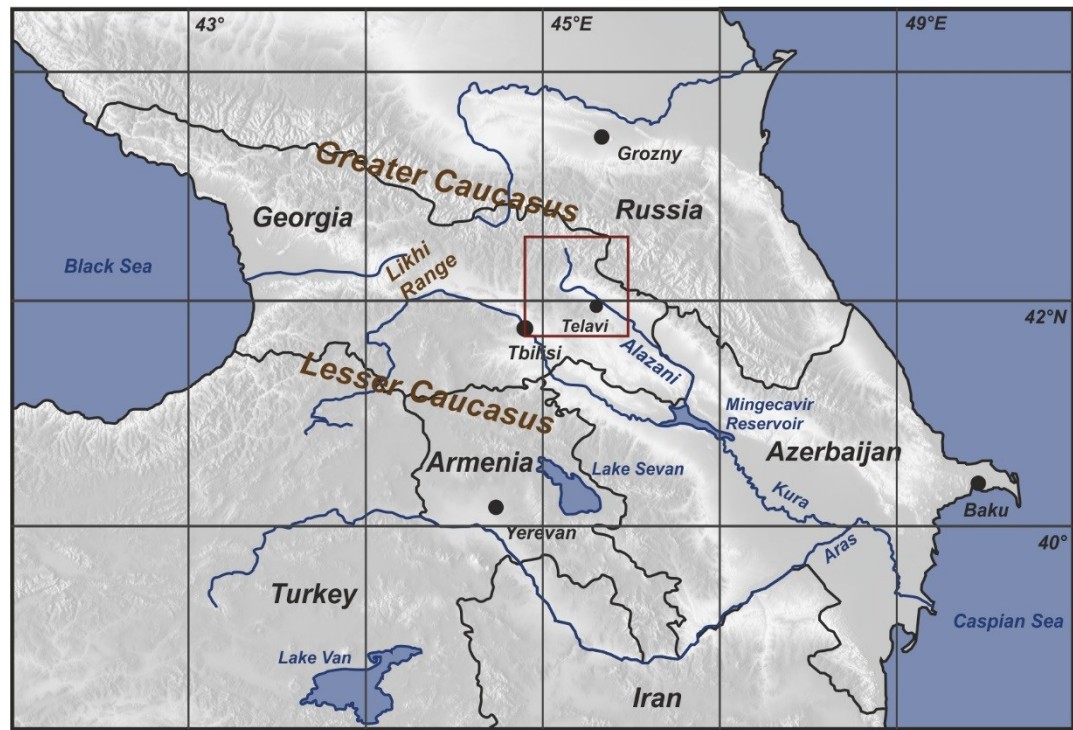

**Figure 1: Overview of the Caucasus Region. The red rectangle marks the study area.**

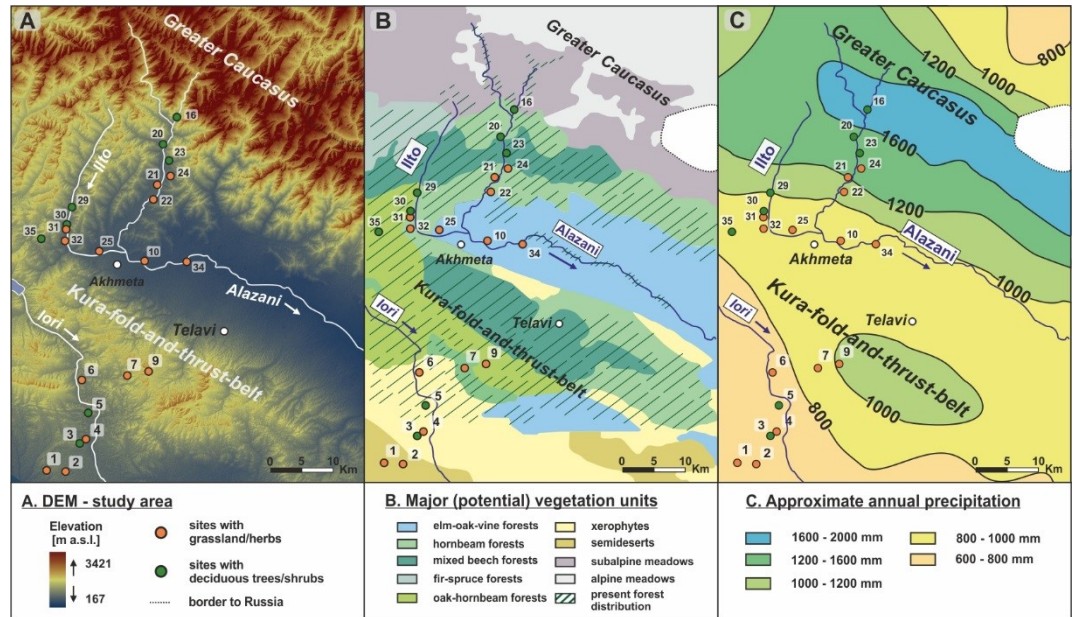

**Figure 2: A.** Digital elevation model of the study area (Aster DEM). Locations of plant and topsoil samples are indicated by an orange dot for sites with grassland/herbs, and a green dot for sites with deciduous trees/shrubs. **B.** Current natural potential vegetation in the study area (Connor & Kvavadze 2008). The current distribution of forests in the study area (second half of 20th century) is derived from Soviet military maps 1:200,000. **C.** Approximate annual precipitation in the study area (based on unpublished precipitation map of the V. Bagrationi Geographical Institute Tbilisi that averages precipitation data between 1960 and 1990).

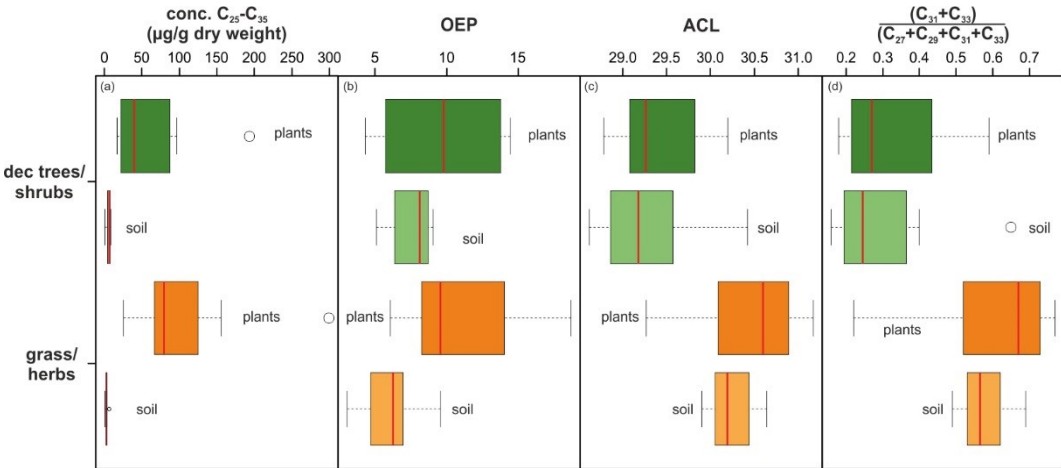

**Figure 3:** Box plots of *n*-alkanes from modern plant and topsoil samples. (a) Concentration ug/g ($C_{25}$-$C_{35}$), (b) Odd-over-even predominance (OEP), (c) Average chain length (ACL), (d) *n*-alkane ratio ($C_{31}$+$C_{33}$)/($C_{27}$+$C_{29}$+$C_{31}$+$C_{33}$). Dec = sites with deciduous trees/shrubs (n = 16); grass = sites with grassland/herbs (n = 28). The box plots show median (red line), interquartile range (IQR) with upper (75 %) and lower (25 %) quartiles (full rectangles) and outliers (dashed lines).

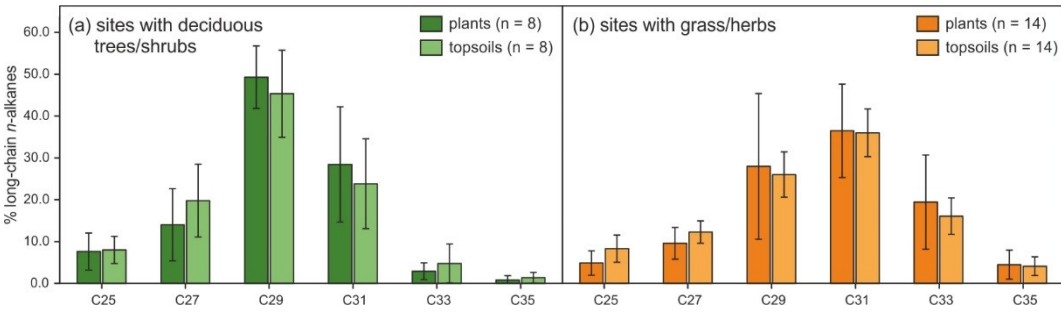

**Figure 4: Chain length distribution patterns for long-chain *n*-alkanes in modern plants and topsoils from sites with (a) deciduous trees/shrubs, and (b) grassland/herbs.**

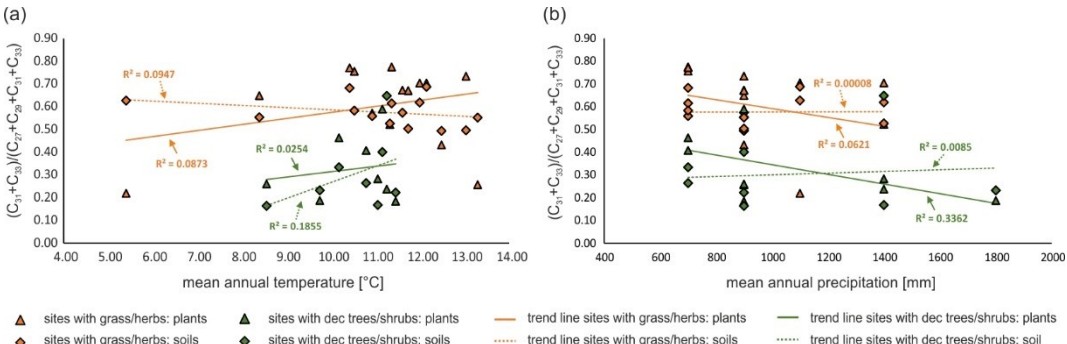

**Fig. 5: *n*-alkane ratios from modern plant and topsoil samples plotted against (a) mean annual temperatures and (b) mean annual precipitation. Dotted trend lines are linear regression lines for *n*-alkanes of modern plants from sites with grassland/herbs (orange) and sites with deciduous trees/shrubs (green), whereas solid trend lines refer to *n*-alkanes from topsoils.**

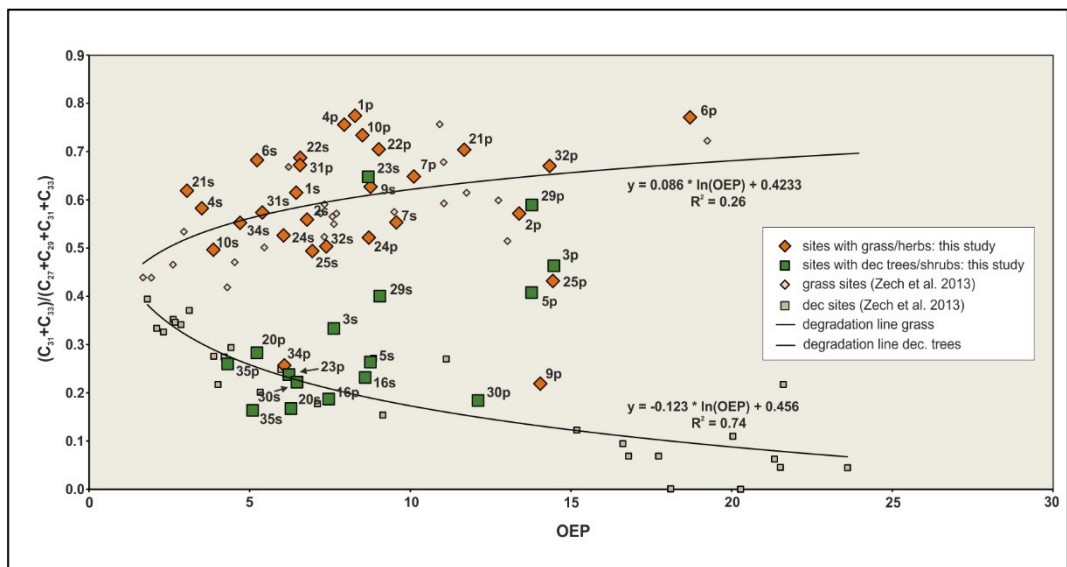

**Figure 6: End-member plot for the *n*-alkane ratio (C₃₁+C₃₃)/(C₂₇+C₂₉+C₃₁+C₃₃). *n*-alkane results from plant and topsoil samples from eastern Georgia were integrated into a dataset from recent grassland and forest sites from Central and South-Eastern Europe (R. Zech et al., 2013). Degradation line for grass/herbs: y = 0.086 * ln(OEP) + 0.4233. Degradation line for deciduous trees/shrubs: y = -0.123 * ln(OEP) + 0.456.**