# Peer review of "Leaf wax *n*-alkanes in modern plants and topsoils from eastern Georgia (Caucasus) – implications for reconstructing regional paleovegetation"

_Biogeosciences, 2017_

## Short Comment (SC1) · 25 Jul 2017

I would like to refer the authors to our manuscript (Holtvoeth et al., Biogeosciences, 13, 795-816, 2016) wherein we report the distributions of n-alkyl compounds from leaf litter and soils of the Ohrid Basin (Western Balkans). Similar to the authors we observe a bimodal n-alkane distribution in the topsoil from a beech forest, with the grass-derived C31 n-alkane dominating over the beech litter-derived C27 n-alkane.

Apart from a change in vegetation cover, another explanation for this observation would

be that n-alkanes from grasses overproportionally enter the soil lipid pool. In our study, the grasses contained about 4-times the amount of n-alkanes (32 vs. 142 microgram/gram dry weight). Furthermore, the autumnal grass biomass is less mobile than leaf litter and remains in-situ. Thus, the proportion of n-alkanes from the grassy undergrowth contributing the soil lipids is almost certainly higher than that from leaf litter. The amount of grass growing even underneath a closed deciduous canopy can vary greatly depending on a range of local factors such as substrate quality, soil moisture, slope and direction of slope etc., which may complicate the interpretation of n-alkane distributions in environmental archives. Generally, though, I agree with the authors that n-alkane distributions can provide useful paleoenvironmental proxies if calibrated locally, and I strongly support the approach to investigate the modern end-member lipid sources and pools, i.e. plant matter and soils, for such calibration.

With best wishes,

Jens Holtvoeth

---

## Referee Comment (RC1) · Anonymous Referee #1 · 25 Sep 2017

Bliedtner et al. report their results of leaf wax n-alkane compositions in eastern Georgia in the central southern Caucasus region covering grassland and forests. They found that n-alkanes are valid chemotaxonomic indicators to differentiate grasses from trees, with the former containing more longer homologues than the latter. This result does not surprise me although it may not be true everywhere. Nevertheless, I am still not sure if it can be used definitely as a paleovegetation indicator because they did not provide detailed environmental parameters, and numerous studies also tend to attribute changes in n-alkane chain length to climate change. This is a dilemma because vegetation may shift with climate change. So what I am interest in is whether the authors can discriminate the separate roles of vegetation and climate factors in modulating n-alkane compositions. So I would like the authors provide more related data and discuss more on them.

1. A more detailed dataset of environmental and climatic parameters, such as temperature, humidity (aridity), precipitation, etc., along the sampling transect, or even for each sampling site, is needed. These data should be examined to see whether and how they influence the n-alkane compositions in general. Authors are encouraged to discuss separate roles of vegetation and environment in modulating n-alkane distributions. For example, as noted in the text, samples 3p, 9p, 25p, 29p, 34p do not show composition patterns as expected. The authors think that these samples may have been influenced by climate. However, the interpretation is rather qualitative and unclear. I guess some plant samples of the same species may have been distributed along climatic gradients. If so, data of these samples are valuable and should be sorted out to see their possible responses to climatic change.

2. The degradation lines in figure 5 are interesting. But it is obvious that the data are much scattering. I would like to see more discussion on the causes of the scattering, including, e.g., climatic factors, disproportional input of leaf waxes to soils from different plants. Also, if the causes are significant, the authors should admit the weakness of the end-member model.

3. As the authors stated in the text, this study is region specific and results appear different from other regions and the globe. It is expected that a comparison of this work with others, and hence a more comprehensive study may improve this paper and is greatly helpful for readers. I suggest the authors give a try.

---

## Referee Comment (RC2) · Anonymous Referee #2 · 4 Feb 2018

This manuscript talked about the distribution pattern of the leaf wax n-alkanes in modern plants and topsoils from eastern Georgia (Caucasus), potentially could be used the related parameters for regional paleoenvironmental reconstructions. The established parameters focused on the C27+C29+C31+C33. The analytical method including the pretreatment and GC-FID measurement is simple and practical. The logic is from plant to topsoil which is trying to finally locate to the regional sediment archives. The result in Fig.5 seems good since higher n-alkane ratios (C31+33/C27+29+31+33) appeared in the grassland and lower ratios occurred in the deciduous sites, , which is consistent

with the fact from modern plant.

1. In Fig.3, larger differences seem appeared in the ACL (why not used) whilst equal values occurred in OEP. Obviously, the ACL values were not used. Its relationship between ACL and n-alkane ratios should be provided in Fig.5, as least provide in the supplementary materials.

2. In Fig. 4, it seems that similar distribution occurred on the modern plant and topsoil. For a better effect, grassland and deciduous site are suggested to be classified rather than between plant and topsoils.

3. In Fig.5, the degradation line for grassland seems perfect but is not ok for the deciduous-site if the logic from plant to soil is implemented, which is usually decreases with low OEPs and with a converging degradation line. But it is not the case, such as 35p→35s, 20p→20s,16p→16s, as well as 9p→9s, 23p→23s, 34p→34s. It seems complicated in the deciduous site for the degradation of OM.

4. According to the results, the potential regional paleoenvironmental reconstructions should be limited to the paleovegetation as illustrated in title, this should be careful in the abstract and conclusion part.

---

## Author Comment (AC1) · 24 Feb 2018

Dear Jens Holtvoeth,

Thank you for your interesting and engaging comment on our local leaf wax n-alkane calibration study and the reference towards your investigations in the Ohrid Basin. We will include your observations on the n-alkane distribution from litter and topsoils under oak and beech forests in our discussion. Interestingly, you observe a C31 dominance in the topsoils of your beech-dominated forest sites without a strong grassy undergrowth,

whereas the beech leaf litter shows a strong dominance of C27. So far, we assumed land-use changes and intensive grazing as possible drivers for the deciduous site of our transect that shows a C31 dominance in the topsoil. Your observation that n-alkanes from grasses enter overrportionally the topsoil, whereas leaf litter is more mobile, is a very interesting explanation and will be included and discussed in the manuscript. Apart from the deciduous site with the C31 dominance in the topsoil, all deciduous sites along our transect seem to incorporate the C29 n-alkane signal from plant/litter into the soil. However, those sites have a closed canopy with less grassy undergrowth, but we agree that the grassy undergrowth depends on a range of local factors that might complicate a clear interpretation of the n-alkane distribution as a paleovegetational proxy.

Best wishes,

Marcel Bliedtner

---

## Author Comment (AC2) · 24 Feb 2018

Dear Referee,

we would like to thank you for reviewing our manuscript and your comments/suggestions. We will revise the manuscript according to your suggestions. Please find below our detailed response:

1. »A more detailed dataset of environmental and climatic parameters, such as temperature, humidity (aridity), precipitation, etc., along the sampling transect, or even

for each sampling site, is needed. These data should be examined to see whether and how they influence the n-alkane compositions in general. Authors are encouraged to discuss separate roles of vegetation and environment in modulating n-alkane distributions. For example, as noted in the text, samples 3p, 9p, 25p, 29p, 34p do not show composition patterns as expected. The authors think that these samples may have been influenced by climate. However, the interpretation is rather qualitative and unclear. I guess some plant samples of the same species may have been distributed along climatic gradients. If so, data of these samples are valuable and should be sorted out to see their possible responses to climatic change.«

We agree that climate can play an important role in the leaf wax distribution. To address those issues, we will correlate environmental data, namely mean monthly temperatures and precipitation from the vegetation period, with our n-alkane distribution from the investigated sites. As no climate stations are available along our investigated transect and for each sampling site at all, we will use the WorldClim – Global Climate Data (Fick and Hijmans, 2017) that provides interpolated average monthly climate data with a spatial resolution of 1 km2. A more comprehensive discussion about the different roles of vegetation and environmental influences on the n-alkane distribution will then be included.

2. »The degradation lines in figure 5 are interesting. But it is obvious that the data are much scattering. I would like to see more discussion on the causes of the scattering, including, e.g., climatic factors, disproportional input of leaf waxes to soils from different plants. Also, if the causes are significant, the authors should admit the weakness of the end-member model.«

Based on our correlation with environmental factors, we will include a more comprehensive discussion about possible climate-induced causes of end-member scattering, as well as species-specific causes and degradation effects.

3. »As the authors stated in the text, this study is region specific and results appear

different from other regions and the globe. It is expected that a comparison of this work with others, and hence a more comprehensive study may improve this paper and is greatly helpful for readers. I suggest the authors give a try.«

A more detailed comparison of our results with other regions will be included.

References:

Fick, Stephen E.; Hijmans, Robert J. (2017): WorldClim 2. New 1-km spatial resolution climate surfaces for global land areas. In: Int. J. Climatol 37 (12), S. 4302–4315. DOI: 10.1002/joc.5086.

---

## Author Comment (AC3) · 24 Feb 2018

Dear Referee,

we would like to thank you for reviewing our manuscript and your comments/suggestions. We will revise the manuscript according to your suggestions. Please find below our detailed response:

1. »In Fig.3, larger differences seem appeared in the ACL (why not used) whilst equal values occurred in OEP. Obviously, the ACL values were not used. Its relationship

between ACL and n-alkane ratios should be provided in Fig.5, as least provide in the supplementary materials.«

Like the n-alkane ratio, the ACL show distinct differences in the n-alkane distribution from grass and deciduous sites and is thus suitable to describe differences in the vegetation composition. We will provide a comparison of both ratios used for the end-member model (i.e. Fig. 5) in the supplementary materials. However, compared to the ACL, the n-alkane ratio shows greater discrimination power in the end-member model between grasses and deciduous trees and seems more suited for describing vegetation differences along our investigated transect.

2. »In Fig. 4, it seems that similar distribution occurred on the modern plant and topsoil. For a better effect, grassland and deciduous site are suggested to be classified rather than between plant and topsoils.«

Will be done so.

3. »In Fig.5, the degradation line for grassland seems perfect but is not ok for the deciduous-site if the logic from plant to soil is implemented, which is usually decreases with low OEPs and with a converging degradation line. But it is not the case, such as 35p!35s, 20p!20s,16p!16s, as well as 9p!9s, 23p!23s, 34p!34s. It seems complicated in the deciduous site for the degradation of OM.«

We agree that OEP values should decrease from plants to subsoils, i.e. with ongoing degradation. Up to now, we do not fully understand the degradation of leaf wax n-alkanes in soils, but we will include a more comprehensive discussion about degradation effects.

4. »According to the results, the potential regional paleoenvironmental reconstructions should be limited to the paleovegetation as illustrated in title, this should be careful in the abstract and conclusion part.«

Will be done so.

---

## Author Response (AR2)

Dear Editor,

we would like to thank you for your effort with our manuscript and the constructive comments/suggestions of both reviewers. We have revised the manuscript according to the reviewer suggestions. Please find below our point-by-point reply and the manuscript with marked changes.

With best wishes on behalf of the co-authors,

Marcel Bliedtner

Anonymous Reviewer #1

1. >>A more detailed dataset of environmental and climatic parameters, such as temperature, humidity (aridity), precipitation, etc., along the sampling transect, or even for each sampling site, is needed. These data should be examined to see whether and how they influence the n-alkane compositions in general. Authors are encouraged to discuss separate roles of vegetation and environment in modulating n-alkane distributions. For example, as noted in the text, samples 3p, 9p, 25p, 29p, 34p do not show composition patterns as expected. The authors think that these samples may have been influenced by climate. However, the interpretation is rather qualitative and unclear. I guess some plant samples of the same species may have been distributed along climatic gradients. If so, data of these samples are valuable and should be sorted out to see their possible responses to climatic change.<<

   We agree that climate can play an important role in the leaf wax distribution. To address those issues, we will correlate environmental data, namely mean annual temperatures and precipitation, with our n-alkane distribution from the investigated sites. However, climate data for eastern Georgia are scarce because only few climate stations are located within the study area, and almost all stations show large data gaps after 1990. So, reliable climate data cannot be derived from climate stations along the investigated transect or even for each sampled site. We therefore used mean annual temperature data of the climate station Tbilisi that is located at 490 m a.s.l.. Based on this reference station, mean annual temperatures for our investigated sites were calculated under the assumption that temperature declines linearly with 0.65 k by 100 m increase in elevation. For precipitation, we used an unpublished precipitation map of Georgia (V. Bagrationi Geographical Institute Tbilisi). We also tried the WorldClim – Global Climate Data (Fick and Hijmans, 2017) that provides interpolated average monthly climate data with a spatial resolution of 1 km$^2$. However, robust temperature and precipitation data could not be derived for eastern Georgia from that dataset, most likely due to the coarse spatial resolution.

2. >>The degradation lines in figure 5 are interesting. But it is obvious that the data are much scattering. I would like to see more discussion on the causes of the scattering, including, e.g., climatic factors, disproportional input of leaf waxes to soils from different plants. Also, if the causes are significant, the authors should admit the weakness of the end-member model.<<

   Based on our correlation with environmental factors, we will include a more comprehensive discussion about possible climate-induced causes of end-member scattering, as well as species-specific causes and degradation effects.

3. >>As the authors stated in the text, this study is region specific and results appear different from other regions and the globe. It is expected that a comparison of this work with others, and hence a more comprehensive study may improve this paper and is greatly helpful for readers. I suggest the authors give a try.<<

   A more detailed comparison of our results with other regions will be included.

Some technical corrections:

1. L16-17, P1. "temperate and tropical regions" can also be "semi-humid to semi-arid". Rephrase it.

- Will be rephrased.

2. L13, P4. "C25 and C35"? I guess "and" should be "to".

   - Will be done so.

3. L16 &18, P5. ACL and OEP have been explained in previous text, so no need of full names.

   - Will be done so.

4. L5, P6. "What" seems incorrect, maybe "which".

   - Will be done so.

   Will be done so.

[revised manuscript text omitted]